# Review of Methodological Approaches to Human Milk Small Extracellular Vesicle Proteomics

**DOI:** 10.3390/biom11060833

**Published:** 2021-06-03

**Authors:** Brett Vahkal, Jamie Kraft, Emanuela Ferretti, Minyoung Chung, Jean-François Beaulieu, Illimar Altosaar

**Affiliations:** 1Department of Biochemistry, Microbiology and Immunology, University of Ottawa, Ottawa, ON K1H 8L1, Canada; bvahkal@uottawa.ca; 2Department of Molecular and Clinical Medicine, University of Gothenburg, SU Sahlgrenska, 41345 Gothenburg, Sweden; jamie.kraft@wlab.gu.se; 3Division of Neonatology, Department of Pediatrics, Children’s Hospital of Eastern Ontario, Ottawa, ON K1H 8L1, Canada; eferretti@toh.ca; 4Department of Molecular Genetics, University of Toronto, Toronto, ON M5S 1A8, Canada; minyoung.chung515@gmail.com; 5Department of Immunology and Cell Biology, Université de Sherbrooke, Sherbrooke, QC J1E 4K8, Canada; jean-Francois.Beaulieu@USherbrooke.ca

**Keywords:** extracellular vesicles, exosomes, human milk, mass spectrometry, proteomics

## Abstract

Proteomics can map extracellular vesicles (EVs), including exosomes, across disease states between organisms and cell types. Due to the diverse origin and cargo of EVs, tailoring methodological and analytical techniques can support the reproducibility of results. Proteomics scans are sensitive to in-sample contaminants, which can be retained during EV isolation procedures. Contaminants can also arise from the biological origin of exosomes, such as the lipid-rich environment in human milk. Human milk (HM) EVs and exosomes are emerging as a research interest in health and disease, though the experimental characterization and functional assays remain varied. Past studies of HM EV proteomes have used data-dependent acquisition methods for protein detection, however, improvements in data independent acquisition could allow for previously undetected EV proteins to be identified by mass spectrometry. Depending on the research question, only a specific population of proteins can be compared and measured using isotope and other labelling techniques. In this review, we summarize published HM EV proteomics protocols and suggest a methodological workflow with the end-goal of effective and reproducible analysis of human milk EV proteomes.

## 1. Introduction

Human milk (HM) is the optimal nutrient source to promote growth of infant via a wide range of macro- and micronutrients [1,2,3,4,5,6,7]. Characterization of the HM proteome has contributed greatly to our understanding of the dynamic fluid and its wide range of additional biological properties. The bioactive components of HM can facilitate antimicrobial protection and immune modulation through secretory antibodies, oligosaccharides, lactoferrin, cytokines, proteins, and the recently characterized–extracellular vesicles (EVs) [2,8,9,10,11].

Formula milk feeding is necessary in situations where HM may not always be available, such as in neonatal intensive care units [12,13]. Significantly, formula milk is deficient in several of the bioactive components found in HM [14,15], including lipids [16]. For example, alkylglycerol-type ether lipids are absent in formula milk but are important in the development of healthy adipose tissue via immune-metabolic signaling [17]. MicroRNAs are also lacking in formula milk [18,19,20,21,22], which are an abundant cargo of HM EVs with an increasingly evident role in infant immune development [23,24,25]. Notably, vesicles detected in formula milk appear to not have EV markers, and are compromised morphologically [22,26], wherein casein micelles have been identified within the population instead [22]. Thus, intact EVs are an important component of HM. The identification of immunomodulatory proteins in HM EVs [11] indicate that further proteomics analyses and characterization of EVs can inform on components to include in formula to help infants thrive [27,28,29,30,31,32].

The proposed mediators for downstream bioactivity, HM EVs are surrounded by a lipid bilayer, secreted by nearly all cells, and found across biological material [11,33,34,35,36]. They contain proteins, RNA and DNA, which can convey cell-to-cell signaling [37,38]. Among the subclasses of EVs are microvesicles and exosomes [39], with distinct surface markers [39,40]. As a subset of EVs, exosomes were first described in 1983 [41,42], but characterized in 2016 by Kowal et al. with surface markers of CD9, CD63 and CD81 [43]. They are considered among a small class of EVs, ranging in size from 20–200 nm [36,44]. Due to limitations in EV isolation procedures, the denotation of “exosomes” to a heterogenous EV population obtained from biofluids has recently been discouraged. This update has been extensively covered in the current guideline for EV research (MISEV 2018) [45]. In this review, we use the term “EVs” or “small EVs”, in alignment with MISEV standards [45], to encompass proteomic studies of HM vesicles, where isolation method yielded a population in the average size of <200 nm.

EVs are proposed to convey signals systemically and between cells [46,47,48]. In a mouse study, fluorescently labeled bovine milk EV miRs were bioavailable following oral intake and distributed to intestinal mucosa, spleen, liver, heart and brain [46]. On a cellular level, EVs from milk have been shown to enter intestinal cells [23,49], macrophages [50,51], peripheral blood mononuclear cells [52], and vascular endothelial cells [53]. Indeed, HM EVs can withstand digestion, indicating delivery of bioactive cargo to the infant gut lumen [54,55].

Relevant for infant health, functional studies have demonstrated that HM EVs significantly increase cell proliferation [56,57,58]. Three independent studies of rat necrotizing enterocolitis (NEC), an inflammatory gut disease, showed reduction in NEC severity following feeding or IP administration of HM EVs in vivo [30,31,56]. In a gingival epithelial cell line, re-epithelization of cells in the presence of HM EVs was shown to be p38 MAPK-dependent, and cell adhesion molecules E-cadherin and EPCAM were downregulated [59]. In immune regulation, CD4 + T-cell activation and regulatory T-cell induction are inhibited in response to treatment with HM EVs [59]. Treatment also dampened agonist-induced response of toll-like receptors (TLR) 3 and 9, pro-inflammatory cytokines IL-6 and CXCL8 were downregulated [59]. In peripheral blood mononuclear cells, treatment with HM EVs inhibited release of IL-2, IFNγ, TNFα, and promoted production of IL-5 [11].

Despite emerging functional data, much of our current understanding of pathways affected by HM cargo is reliant on big data sets derived from “EV-omics”, including proteomics [36]. At this point, we note that current proteomics is evolving in a direction of increased effectiveness of methods through miniaturization [60,61,62,63] and automation [64]. In addition to advancing the field of single-cell and spatial proteomics, sample preparation time, sample quantity and reagents are also reduced, and peptide analysis is faster [62,64,65]. Here, we suggest the use of nano-liquid chromatography and tandem mass spectrometry for reliable HM EV proteomics, which utilizes several of the aforementioned improvements in proteomics workflow [62,66].

HM is a complicated fluid, which extends to its EV cargo. EV isolation and proteomics methods are ever evolving to further distinguish EV populations and signatures from originating cells [67]. Existing literature is exploring this in plasma and cell-culture derived EVs, which can also inform the future studies of HM EVs. The use of proteomics can reveal functional data related to identified proteins using large-scale analyses, wherein complex molecular mechanisms and pathways are an important focus. In functional analyses, understanding the effects of HM EV treatment is enriched by pathway analyses of omics cargo to postulate signaling pathways that lead to the observed effect. Several studies have utilized this approach thus far [11,56,59], which can be helpful for additional verification experiments, future directions.

The diverse approaches and areas of HM EV study have led to the expansion of the field, where a five-fold increase is seen in the last eight years in publications with the title “exosom * or vesicle *”, though publications on HM exclusively are in the minority (Figure 1A, Web of Science analytics, 21 April 2021). An upward trend is observed in published works on milk EVs, exosomes, and their investigation using mass spectrometry, however, only a quarter of those studies examine HM (Figure 1B). In this review, we summarize existing literature on HM EV proteomics and suggest a reliable method for EV protein isolation, peptide preparation, detection and bioinformatic annotation. We include alternative methods from EV studies of other biofluids, which may enhance the functional interpretation of HM EV proteome.

## 2. Human Milk Small Extracellular Vesicle Isolation

HM is a biofluid unique in its composition, with a high concentration of proteins and lipids [68]. Therefore, considerations must be taken in EV isolation to separate the contaminants prior to omics analyses. The experiments conducted on HM EVs vary in their approaches of sample preparation for proteomic or other experimental analyses (Table 1). For the majority of HM EV isolations, a series of differential and ultra-centrifugation steps are conducted to pellet small EVs or exosomes [28,56,69,70,71]. Several published protocols have combined differential centrifugation with a sucrose gradient [11,72,73,74]. While centrifugation and gradient isolation techniques can be considered the classic approaches, they are time-consuming and have multiple experimental steps [36]. To make EV isolation technically more accessible, alternative isolation methods are emerging. Size exclusion chromatography columns (e.g., qEV) [75] can yield EVs of superior purity and is increasingly used in the field of non-HM EV isolations [76,77,78,79,80]. Kits (e.g., ExoQuick-TC, PureExo) and precipitation solutions (e.g., total exosome isolation reagent) can provide rapid EV isolation, but result in a higher concentration of contaminants [78,81]. Newer approaches in the field include asymmetric tangential flow filtration, including asymmetrical flow field flow fractionation (AF4), ion-exchange, electrophoresis and dielectrophoresis, for an advanced separation of vesicles [67,78]. AF4, for example, can provide rapid separation of vesicles in up to 1 nm increments [82], but can have a reduced overall yield [67]. Combinations of the above methods are also proving useful in enriching EV populations from complex biofluids, such as dual-mode chromatography (DMC), which has been tested in plasma to remove contaminating lipoprotein particles [83].

Though promising, several of these methods have not yet been established for separation of lipid and protein content when processing HM samples, wherein differential centrifugation of sample may still be necessary to reach desired purity of EVs from milk [75]. The commonly used and established protocol remains a series of differential centrifugations of HM to remove fat and cellular debris, and ultra-centrifugation of the resulting supernatant twice at 100,000× *g* for 1.5 h at 4 °C to obtain a pellet of small EVs [28,56,69,70,71,75]. Methods such as DMC may prove useful in future isolations of HM EVs, especially when combined with differential centrifugation for obtaining a skim milk sample for further processing. Similarly, SEC could be applied instead of ultra-centrifugation to increase the purity of obtained EV sample. However, these approaches need to be validated in HM biofluid prior to use, where it would be informative to compare EV integrity, yield, sample purity to the classical differential and ultra-centrifugation method.

A question whether the HM protein and lipid contents are sufficiently removed from EVs, remains. Biological variations exist between mammals, where elephant seal milk contains over 50% fat [84], bovine milk has roughly 4.2% fat [85], and human milk has varying concentrations between 3–5% [86]. Additional contaminants in milk are casein and whey proteins [87], wherein casein is present at a higher concentration in bovine milk versus HM (3–5 versus 26 g/L) [68]. Previous proteomics studies from HM EVs have not reported significant interruptions in proteomics analyses from casein and whey, though it may be a consideration for functional analyses. Following isolation, specific clean-up steps should therefore be taken to ensure sensitive and accurate mass spectrometry analyses. These purification steps can include precipitation, which can utilize acetone or methanol among other reagents, or extraction of the protein using solid phase (e.g., C18) and can remove impurities prior to sample digestion [88].

Prior to further sample preparation for proteomics, it is essential to confirm presence, size distribution and concentration of EVs in purified sample [45]. This should include confirmation of surface markers [39,45]. For small EV populations, they should include CD9/63/81, and/or Annexin A1 [39,45]. Relying on morphology, scanning electron microscopy can be used to identify size and distribution of EVs. Nanoparticle tracking analysis can be used to map vesicle sizes in a sample and to confirm presence of 20–200 nm vesicles [35,39,70,89,90].

**Table 1 biomolecules-11-00833-t001:** Summary of experimental approaches for small extracellular vesicle (EV) isolation from human milk.

Reference	EV Isolation	EV Experiment
[11]	1. Differential centrifugation 2. Filtration 3. Differential and ultra-centrifugation 4. Sucrose gradient	LC-MS/MS
[72]	1. Differential centrifugation 2. Filtration 3. Ultracentrifugation 4. Sucrose gradient 5. Ultra-centrifugation	Characterization assays
[91]	1. Differential centrifugation 2. ExoQuick Exosome Isolation Kit	Cell culture treatments
[73]	1. Differential centrifugation 2. Sucrose gradient 3. Ultra-centrifugation	Proteomic characterization of vesicles
[74]	1. Differential and ultra-centrifugation 2. Filtration 3. Sucrose gradient 4. Filtration	iTRAQ-coupled LC-MS/MS
[49]	1. Differential centrifugation 2. Filtration 3. ExoQuick-TC solution kit	Cell culture treatments
[92]	1. Differential centrifugation 2. Filtration 3. ExoQuick Exosome Precipitation Solution 4. Centrifugation	miR analysis
[28,56]	1. Differential centrifugation 2. Filtration 3. Ultra-centrifugation	Cell culture treatments; iTRAQ-coupled LC-(ESI)-MS/MS
[55]	1. Differential centrifugation 2. Filtration 3. ExoQuick-TC solution kit	Next-generation sequencing, in vitro assays
[31]	1. Differential and ultra-centrifugation	Cell culture, in vivo treatments

## 3. Preparation of Human Milk Extracellular Vesicles for Mass Spectrometry

To date, few protocols are available for the preparation of HM EVs for liquid chromatography and mass spectrometry [11,56,73,74]. Bottom-up proteomics are commonly used for EV analyses, where all proteins are first digested and then peptides are separated prior to analysis in a mass spectrometer [93,94,95]. Specific to sample preparation, trypsin digestion optimization is needed to obtain maximal digestion of arginyl and lysyl peptide bonds in human milk exosome proteins.

EV sample preparation can include a variety of surfactants, e.g., SDS, n-dodecyl β- D-maltoside (DDM), and Triton X-100, which are used to prepare proteins with varying efficiency, depending on the sample type, for trypsin digestion [96,97,98,99]. Prior studies on human milk exosomes utilized different concentrations of sodium dodecyl sulfate in a denaturing buffer with Tris-HCl and urea, yielding reliable analysis [11,74]. Following digestion, de-salting of the sample avoids damage to the liquid chromatography and mass spectrometry equipment. There are multiple kits available for sample cleanup, e.g., ZipTips [100]. Either prior to or following digestion, peptides can be fractionated using chromatography techniques before running the milk-derived EV sample in a tandem mass spectrometer. This can be beneficial in order to separate peptides and can help better characterize the diverse exosomal protein content [88].

Multiple fractionation approaches have been used to separate exosomal peptides, which generally include a two-step separation. For human milk exosome samples, strong cation exchange chromatography (SCX) coupled with reverse phase was used by Admyre et al. (2007), as well as Yang et al. (2017) (Appendix A). In Van Herwijnen et al. (2016)*,* in-gel digestion and separation were performed. In all cases, second phase separation utilized nano-liquid chromatography instrumentation. Proteomics studies on EVs from other biological sources have also included gel separation of intact proteins prior to digestion, followed by reverse phase separation [95], or two reverse phase separations with varied conditions [94]. The primary separation step might be omitted for proteomics characterization. For this one step approach it would be worthwhile to consider whether: (1) sufficiently pure EV sample is obtained; (2) optimal trypsin digestion conditions are developed; and (3) high performance or nano-liquid chromatography is used coupled with high-resolution tandem mass spectrometry for peptide separation and analysis.

## 4. Mass Spectrometry and Proteomics of Human Milk Extracellular Vesicles

The use of the tandem mass spectrometer allows distinct selection of precursor and fragmentation ions for detection, obtaining high confidence protein matches [101]. Admyre et al. in 2007 used QSTAR Pulsar mass spectrometer, which couples to both matrix-assisted laser desorption ionization (MALDI) and electrospray ionization (ESI), suitable for sequencing proteins in a wide mass range. Both Van Herwijen et al. (2016) and Yang et al. (2017) used a Q-Exactive instrument, where precursor ions were acquired using orbitrap, and ten most intense precursors were sent to fragmentation using high-energy C-trap dissociation (HCD). Use of HCD can increase the number of identified peptides, which is an advantage of the Q-Exactive instrument [102]. The most recent study by Wang et al. (2019) utilized Triple TOF 5600 Plus, which is a triple quadrupole coupled with time of flight mass analyzer and allows for a variety of mass scans [103].

Depending on the research question, mass spectrometry spectra can be generated and analyzed based on data-dependent acquisition (DDA) or data-independent acquisition (DIA) for global characterization of samples, or using multiple reaction monitoring (MRM) for specific EV components, e.g., for distinct cell membrane proteins [104]. In many current HM EV studies, DDA is used in tandem mass spectrometry, where selected high abundance precursor ions from MS1 are sent for fragmentation and detection in an MS2 scan [11,56,88]. The resulting spectra are queried in databases, using searching algorithms such as Sequest, Mascot, X!Tandem and/or Andromeda/MaxQuant, for matches and generation of the corresponding derived protein list [88]. In the 2007 study by Admyre et al., the authors identified over 70 proteins using a label-free technique; in comparison to Yang et al. (2017), where the authors reported over 900 proteins from bovine and human milk exosomes using iTRAQ labeling [36].

New strategies have been developed for DIA, e.g., sequential window acquisition of all theoretical fragment ion spectra (SWATH) and MS^E^ [88,105,106,107]. They allow selection of the majority of precursor ions in a selected window of the MS spectrum to be sent on to fragmentation simultaneously [107,108]. This method could identify low abundance proteins, which would otherwise remain undetected or characterized as noise in MS spectra [107]. In small EV characterization, it would be beneficial to identify proteins of lower abundance [105]. Two recent studies of urine extracellular vesicles and exosomes utilized MS^E^ and SWATH, which allowed high confidence identification of 888 and 1877 proteins respectively at less than 1% false discovery rate for the former [109,110]. However, these DIA methodologies are still being developed and remain untested on milk EVs.

For quantitative and comparative proteomics, both label-free and isobaric labeling methods have been developed and reported. Label-free techniques can be beneficial and cost-effective for a more thorough characterization of EVs. However, they are more sensitive to differences in sample preparation and external factors when compared to isobaric labeling [101,105]. When sample comparisons are required, e.g., differences in term or preterm HM, a labeling technique is useful to compare and quantify the number and concentration of proteins in distinct sample populations [56]. To date, four studies have specifically characterized human milk EVs using data dependent acquisition method with or without label-free or isobaric labeling for quantitative analysis and sample comparisons (Appendix A) [11,56,73,74]. Isobaric labeling has been informative in studies of bovine- and human-milk derived EVs, which determined that EVs differ depending on biological source and disease state [20,72,74]. These findings have raised interest in milk biomarkers to advise on biological status and can rely on the aforementioned proteomics approaches for identification.

## 5. Digestion Optimization for Label-Free Mass Spectrometry of Human Milk Extracellular Vesicles

We performed protein solubilization optimization on term HM small EV samples to test seven detergents and determine the optimal milieu allowing maximal digestion of arginine- and lysine-involving peptide bonds in HM EV proteins. HM was collected from mothers following term birth, wherein colostrum was excluded. Between 15–20 mL of milk was collected per mother and processed immediately. Briefly, whole HM was centrifuged twice at 4600× *g* for 30 min, at room temperature initially and 4 °C for the second centrifugation, to remove fat and cellular debris. The remaining skim layer was centrifuged at 20,000× *g* for 30 min at 4 °C to pellet vesicles larger than 200 nm. The middle layer was then ultra-centrifuged twice at 100,000× *g* for 1.5 h at 4 °C. The resultant EV pellet was resuspended in sterile PBS for storage in −80 °C until experiment.

SDS with and without EDTA at 5 mM, 10 mM and 34 mM concentrations, Digitonin, n-dodecyl β-D-maltoside (DDM), and Triton X-100 were tested as detergents to allow for Trypsin/Lys-C to cleave peptides and be identified with liquid chromatography tandem mass spectrometry (LC-MS/MS). The detergents were added to samples in 1:4 ratio, respectively, and vortexed for five seconds. Following processing, 30 μg of each sample was desalted using TopTips (Fisher Scientific, Waltham, MA, USA). The samples were then eluted three times with 50 μL of 70% acetonitrile, 30% water and 0.1% formic acid and placed in a speed vacuum to dry. The samples were run on the Thermo Scientific^TM^ Orbitrap Fusion^TM^ Tribrid^TM^ (Waltham, MA, USA).

The number of protein IDs identified with high and medium confidence using different detergents DDM, Digitonin, Triton X-100, SDS, SDS 5 mM EDTA, SDS 10 mM EDTA and SDS 34 mM EDTA was 693, 667, 724, 728, 692, 721, 681, respectively (Figure 2). Digesting the EVs with SDS as the detergent provided the highest number of protein IDs (728), as well as 57 unique proteins (Figure 3). However, the most unique proteins (100) were detected using SDS with 34 mM EDTA (Figure 3).

## 6. Bioinformatics Approaches to Human Milk Extracellular Vesicles

Following protein detection, we selected high confidence proteins from all treatments to identify corresponding biological processes using GOrilla [111]. The redundant GO terms were then removed in REVIGO [112] and an updated list of biological processes was obtained. DDM, Digitonin and SDS treatment groups identified the most biological processes (Figure 4). SDS EDTA 34 mM treatment resulted in a narrow biological process identification, though this treatment resulted in identification of most unique proteins (Figure 3). Body fluid secretion and lactation were identified across all treatments (Figure 4), indicating that these processes may be conserved across HM small EVs and serve as a benchmark for sample processing in proteomics.

The number of proteins identified is important for downstream analyses, as well as detection of specific proteins in individual samples. Downstream bioinformatics analyses can provide answers to narrower research questions. To aid such approaches, multiple research groups have submitted their exosome and/or vesicle proteomics results to custom-curated databases, which include ExoCarta [113], Vesiclepedia [114], and EVpedia [115]. EVpedia and ExoCarta were the first available databases for EVs and exosomes [116,117]. EVpedia contains proteins from hundreds of proteome sets obtained from EVs [88]. ExoCarta has a substantial amount of entries from exosomal components, including proteins, lipids, mRNA and miRs, and 81 entries corresponding to human milk [34]. Similar to EVpedia, Vesiclepedia has a wide variety of entries from EVs, including proteins and lipids [34,118]. Therefore, protein matches obtained from database search algorithms (e.g., Mascot) can also be queried in exosomal databases to compare sample variability, disease states and more.

Further enrichment analyses of data can be helpful to identify interacting proteins or networks. Several of such databases are available, including STRING, Profiler, GSEA, Cytoscape and EnrichmentMap [119,120]. Additionally, FunRich, a functional enrichment tool, allows integration with Vesiclepedia, making it a useful tool for EV-omics [121].

## 7. Conclusions

Preparing human milk small EVs for bottom-up proteomics is a multi-factorial process (Figure 5, Appendix A). The first step is sample isolation where the most frequently used technique is differential and ultra-centrifugation, with or without a sucrose gradient. Once small EVs are isolated and confirmed, samples can be prepared for mass spectrometry analysis. Proteins should then be digested and solubilized with a detergent of choice, depending on analysis end-goal and to minimize damage to mass spectrometry equipment. In our experiment, highest number of proteins were detected using SDS with 34 mM EDTA (Figure 2). Peptides should then be separated either in a one or two-step chromatography process, preferably using reverse phase nano-liquid chromatography, followed by analysis in a tandem mass spectrometer, where various high-resolution analyzers have been used in previous studies (Appendix A). If the research question requires sample comparisons or quantification, label-free or isobaric labeling techniques should be used. The latter needs to be added to the sample prior to or following trypsinization. If global protein analysis is the goal, no labeling is required, and data dependent acquisition can be used. Newer approaches in data independent acquisition (e.g., SWATH, MS^E^) can also be utilized [106,109], however, pilot experiments should be conducted for validation of analyses in milk EVs and exosomes.

Human milk small EVs can contribute to our understanding of systemic signaling, nutritional enrichment, immune regulation and much more. EV research relies on accurate proteomics findings to inform on proteins of interest, characterize modes of interactions, and downstream effects of human milk EVs and exosomes. It is essential to obtain reproducible data from human milk small EVs. This bio-analytical field continues to grow, co-evolving with developments in proteomics approaches. The strategies outlined in this review are currently in-use for the intricate process of small EV isolation and analysis, however, new emerging methods will no doubt inform and improve on the processes. For example, the use of data independent acquisition approach may yield even more detailed information about EV-associated proteins.

## Figures and Tables

**Figure 1 biomolecules-11-00833-f001:**
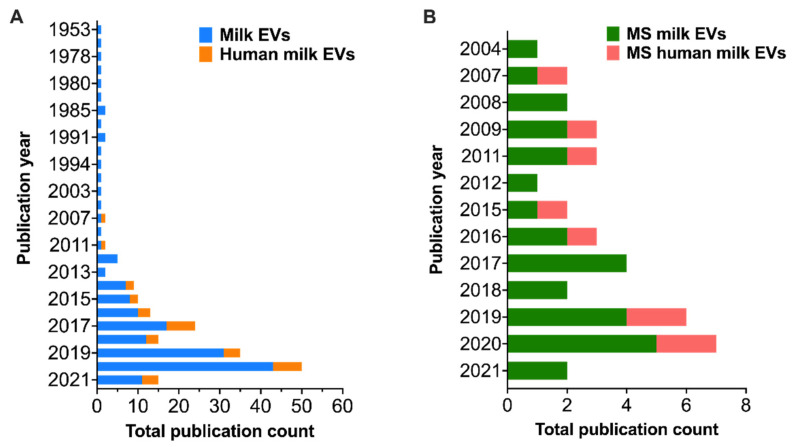
Publication history for extracellular vesicles (EVs) and exosomes. (**A**) Publication count by year for EVs and exosomes in milk (blue) and human milk (orange), where analyses of vesicles in human milk first appear in 2007. (**B**) Publication count by year for mass spectrometry analyses of EVs and exosomes in milk (green) and human milk (pink), where mass spectrometry (MS) analysis of milk vesicles first appeared in 2004. Image generated using Prism 9, data provided by search in Web of Science with combination of terms exosome *, vesicle *, milk, human milk, mass spectrometry, including search constraints of title (**A**) and abstract (**B**), and document type refined by article and review (accessed 21 April 2021).

**Figure 2 biomolecules-11-00833-f002:**
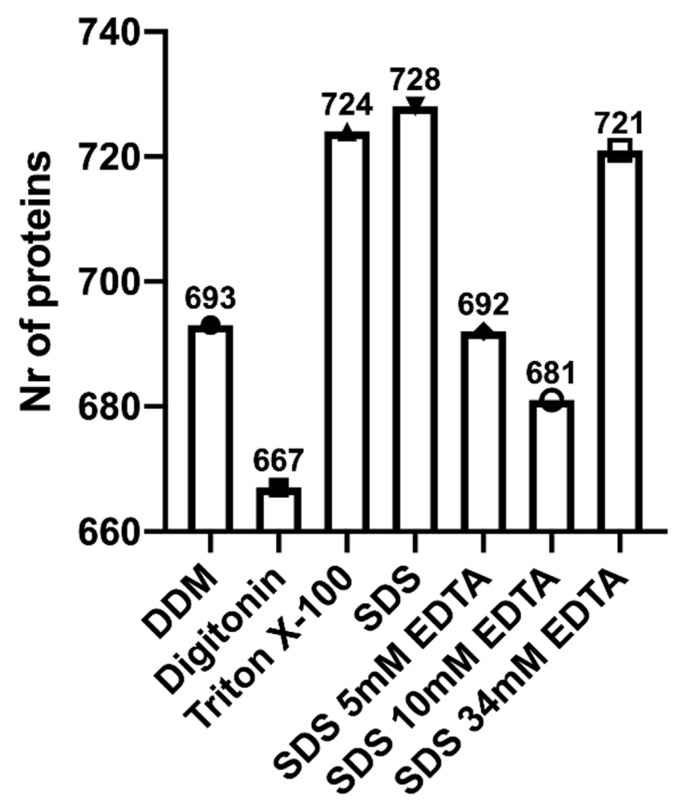
Number of high and medium confidence proteins identified following tandem-MS/MS analysis of human milk extracellular vesicles under seven different denaturing conditions. SDS with and without EDTA at 5 mM, 10 mM and 34 mM concentrations, Digitonin, n-dodecyl β-D-maltoside (DDM), and Triton X-100 were tested as detergents, n = 1. Mass spectrometry analysis from samples processed with each detergent revealed Triton X-100, SDS and SDS with 34 mM of EDTA to be the optimal detergent based on the highest number of proteins identified.

**Figure 3 biomolecules-11-00833-f003:**
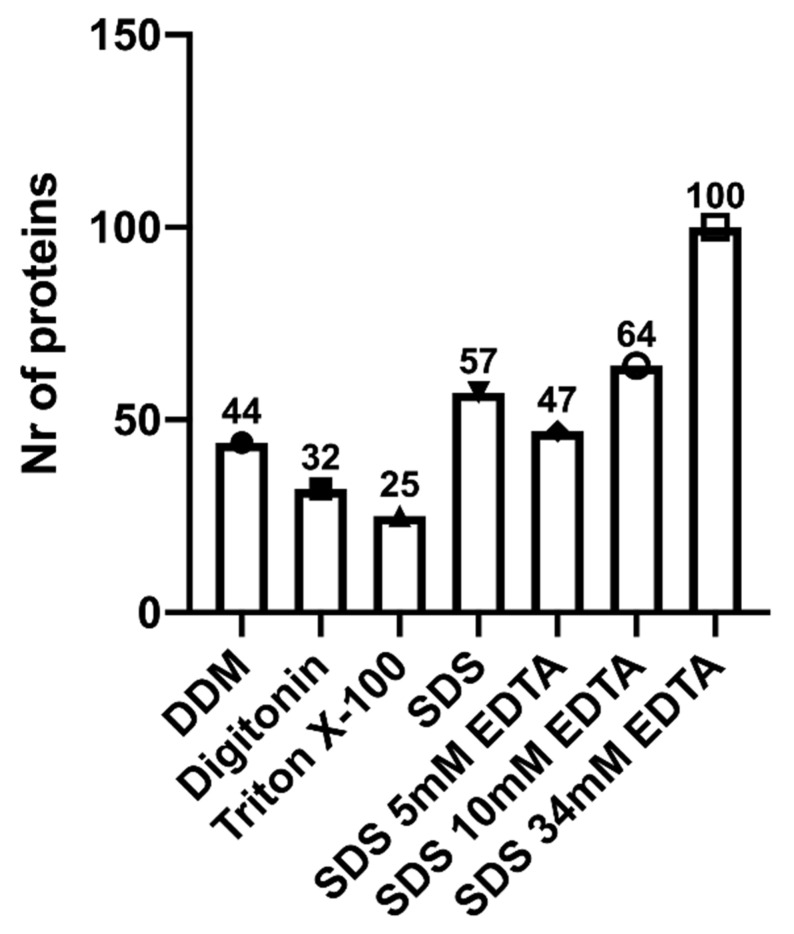
Number of high confidence unique proteins identified following tandem-MS/MS analysis of human milk extracellular vesicles under seven different denaturing conditions. SDS with and without EDTA at 5 mM, 10 mM and 34 mM concentrations, Digitonin, n-dodecyl β-D-maltoside (DDM), and Triton X-100 were tested as detergents, n = 1. Mass spectrometry analysis from samples processed with each detergent revealed SDS with 34 mM of EDTA to be the optimal detergent based on the highest number of unique proteins identified.

**Figure 4 biomolecules-11-00833-f004:**
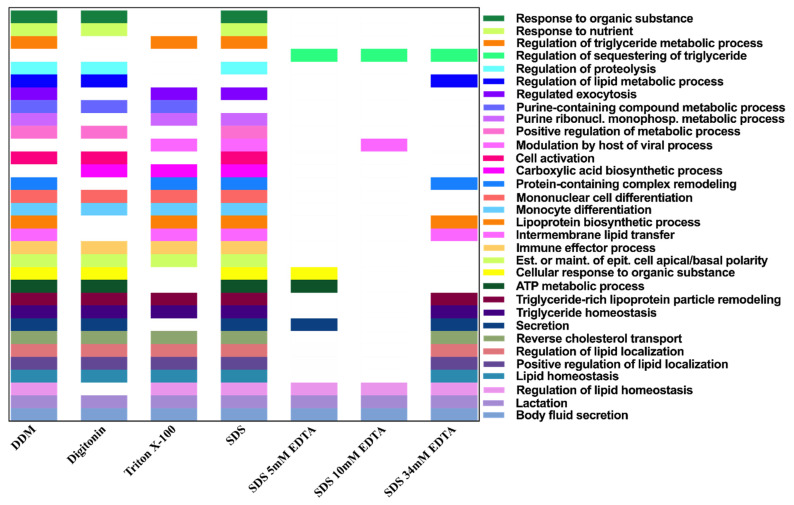
Biological processes corresponding to high confidence proteins detected in human milk extracellular vesicles processed under seven different denaturing conditions. Biological processes that were common to three or more conditions are shown. Highest number of biological processes were identified from human milk extracellular vesicle samples processed with SDS and DDM. Lactation and body fluid secretion were identified across samples under different denaturing conditions. REVIGO analysis of significant GO terms identified by GOrilla analysis, following removal of redundant terms.

**Figure 5 biomolecules-11-00833-f005:**
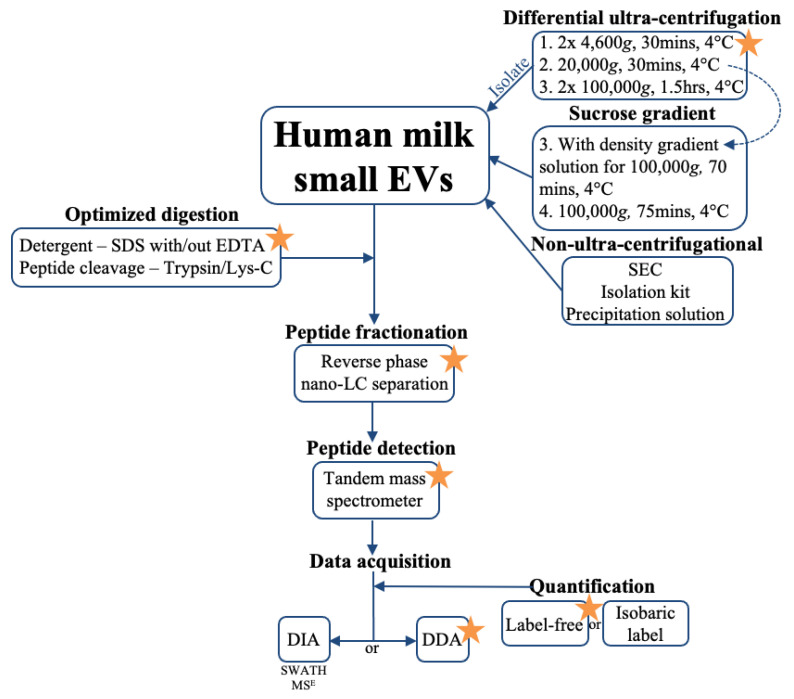
Workflow for human milk small extracellular vesicle (EV) bottom-up proteomics. Human milk small EVs can be isolated using differential centrifugation with or without sucrose gradient, or using non-ultra-centrifugational methods (SEC–size exclusion chromatography). Protein digestion can be achieved using an SDS detergent with or without EDTA, following peptide cleavage with Trypsin/Lys/C, however solubilization agents and trypsinization should be optimized for each experiment depending on analysis end-goals. Chromatographic fractionation of peptides follows, which can include reverse phase nano-liquid chromatography (nano-LC). Peptides are detected by tandem mass spectrometer and data acquired using data independent acquisition (DIA: SWATH-Sequential Windowed Acquisition of All Theoretical Fragment Ion; MS^E^), or data dependent acquisition (DDA), which can be coupled with label-free or isobaric labeling of EV sample. Orange star emphasizes suggested method for use in human milk EV proteomics, as identified in prior publications on human milk EV proteomic analyses.

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
