# Peer review of "Review of Methodological Approaches to Human Milk Small Extracellular Vesicle Proteomics"

_biomolecules, 2021, doi:10.3390/biom11060833_

Round 1

Reviewer 1 Report

In this review, the authors nicely give an overview on proteomics research on human milk EVs. The manuscript is well written and has a good structure. However, I do have some comments about some of the claims made. As a researcher that has 7+ years of experience in the milk EV field and that has done proteomic analysis, I hope the authors will acknowledge my concerns and that we can make this paper even better. As far as the actual mass spectometry, I am not an expert. So about that part of the manuscript I have no detailed comments (I liked reading into it again, and that makes this review a nice and unique addition in the already existing literature).

Page 1:

"Multiple subclasses have been proposed to distinguish 
EVs by their size and surface markers 8,9." The consensus in the EV field is that there is no specific marker or size to distinguish subsets of EVs, as there is always some overlap (I would like to refer to the MISEV guidelines here). The only way to distinguish EV subset is to look at their biogenesis via EM. Thus, despite the references claiming otherwise, this sentence should be removed. 

"Functionally, they are proposed to be a form of extracellular communication for immune system and signal". Recently, we published a paper in the Journal of Extracellular Vesicles (Zonneveld et al.), that actually incorporated proteomics analysis in order to support our findings on the functionality of human milk EVs. It is because of my background as a biologist, I would actually like to have more background into the function of milk and it's EVs in the introduction. Also, what is not discussed in detail (only about the fat/lipids) is that milk is very complex and that this poses a problem for purity and downstream analysis. 

Page 2

About Supl. Table 1, why is Lukasik et al., 2018 in there? Since this is not a proteomics paper and should be excluded from the analysis. 

"Alternative isolation methods are emerging, such as kits (e.g. ExoQuick-TC, Total Exosomes Isolation, PureExo), precipitation solutions (e.g. Total Exosome Isolation Reagent), and size exclusion chromatography 
columns (e.g. qEV)32. These methods can have superior yield and exosome purity but can be more expensive per sample when processing large volumes of HM. The increased EV yield and purity is reported for EV isolation from cell media or blood," Only SEC is considered a reasonable isolation method when considering purity (combined with density gradient separation, it is superior to all other isolation strategies). Precipitation, as done with the commercial kits does not give a pure sample! There are numerous papers that have compared isolation methods (a classical paper is from Van Deun et al., but there are more). This claim really needs to be removed, or otherwise re-written.  

At the end of page 2, the authors discuss that milk fat might be a potential contaminant. I would also like to raise awareness for casein micelles. In our experience (not yet published this, but we are working on a bovine milk EV paper), caseins (from bovine milk especially) pose an ever bigger problem, as these can be co-isolated with EVs (same size, and apparently, also an overlap in density) and influence down-stream analysis, which might include proteomics (we only have western blot data to prove this, but this would then be similar for proteomic analysis). 

"Sucrose from density gradient isolations can also be an 
undesired contaminant." Why? sucrose is known to affect the EV corona (and their functionality), but this wouldn't be a problem for proteomics. Actually, sucrose can be easily diluted and removed from samples. Or, do the authors have specific literature that can back up this claim?

Page 3

"Current minimum requirements include confirmation of surface markers9" I don't get this reference. I would suggest refering to the MISEV2018 guidelines from ISEV. This is a community supported manifest that clearly explains the minimal requirements for reporting on EV research. 

"they may include CD9/63/81 for exosomes, or Annexin A1 for microvesicles" please remove this, as there is no distinctive marker for exosomes or microvesicles. It is also not that important for the manuscript. Most people falsely call their sample exosomes, but the correct term for vesicles isolated from biological fluids or cell culture supernatant is extracellular vesicles (only for those that study biogenesis, the terms are correct and needed). 

"bonds in human milk exosome proteins" Please replace exosome by EV (also in other instances), as this would be correct. 

Page 4

The isolation procedure should be better explained. Give the g-forces, and centrifugation times, etc. Perhaps the authors can look at the EV-TRACK database, or the EV-TRACK paper in Nature. This gives (besides MISEV2018) a nice overview of what other EV researcher need to know in order to understand the methods. Also, the starting volume of milk, and donor information should be given, like lactational stage, etc. 

Page 6

Perhaps interesting to know is that there is software called FunRich that is designed specifically for EV omics analysis. 

Author Response

We thank the reviewer for comments and improvements to manuscript. Please see the attachment with our response.

Reviewer 2 Report

In this work, analytical methods to determine proteins in EVs were overviewed

Proteomic analyses in human milk samples were covered.

REMARKS FOR INITIAL SECTIONS

ABSTRACT (note)

Which era has been covered regarding development of proteomic sample preparation protocols for analysis of human milk?

ABSTRACT (note 2)

Stress the importance to develop methods in this field due to what reasons? List the reasons to remain the progress in this field.

INTRODUCTION (note)

Introduction does not contain information and current status of milk proteomics and position of HM proteome in it. At the end of this section, there are no novelty points that would have confronted the suggested review against previously published overview papers. Please, provide some.

INTRODUCTION (note 2)

Insert some brief description about current status of proteomics developments, i.e. where are the modern trends in proteomics heading to? State-of-the-art of proteomics heads to the higher greenness of the methods in the course of miniaturisation and automation of proteomics and there are also other courses to look for and mention.

This is due to give the junior researcher readers that just have started to deal with proteomics some broaden information.

Here, are some lately published examples worth to mention as for mentioned trends:

https://doi.org/10.1016/j.cbpa.2020.04.018

https://doi.org/10.1016/j.cca.2020.04.015

REMARKS FOR SAMPLE PREPARATION SECTIONS

Draw the proteomic workflow figure to depict the detail of each step involved from sampling EVS to proteomic analysis by LC-MS/MS.

Make a comparison table to confront pros and cons of essential parameters to be optimised during the development of HM proteomic approaches – table will compare different approaches regarding HM proteomics. Regarding supplementary tables, transfer them to the article.

Author Response

We thank the reviewer for their comments and improvements to the manuscript. Please see the attachment for our response.

Round 2

Reviewer 2 Report

Authors have nicely accomplished given remarks. Although, I still consider important to mention the current course of the proteomics in the case of higher greenness of the methods in the course of automation and miniaturisation. The introduction is well done as for description of ECv, however the review is also about proteomics of ECv, thus brief description of proteomics does deserve an attention here. Please, see the down below upgrade of the text content.

INTRODUCTION (upgrade, 6th paragraph)

Generally, proteomics helps to reveal emerging functional data related with analysis of protein compounds at a large-scale level. Also, to comprehend molecular mechanisms and pathways is of prime aim here. At this point, we note that current proteomics heads to the higher effectiveness of the methods in the course of their miniaturization and automation and there are also other state-of-the-art courses of proteomics to look for [https://doi.org/10.1016/j.cbpa.2020.04.018, https://doi.org/10.1016/j.cca.2020.04.015]. Despite emerging of such functional data, much of our current relevant understanding of molecular pathways affect by HM cargo is relianton big data sets derived from „EV-omics“, including mentioned proteomics [35].

Author Response

We thank the reviewer for bringing this to our attention and completely agree that future directions of proteomics is essential to include. We also thank the reviewer for their text suggestion, which was helpful and utilized in our edits. We have incorporated the suggested paragraph (including the references) in paragraph 6, as well as extended paragraph 7.